# Sinulariolide Inhibits Gastric Cancer Cell Migration and Invasion through Downregulation of the EMT Process and Suppression of FAK/PI3K/AKT/mTOR and MAPKs Signaling Pathways

**DOI:** 10.3390/md17120668

**Published:** 2019-11-27

**Authors:** Yu-Jen Wu, Shih-Hsiung Lin, Zhong-Hao Din, Jui-Hsin Su, Chih-I Liu

**Affiliations:** 1Department of Nursing, Meiho University, Pingtung 91202, Taiwan; x00002180@meiho.edu.tw; 2Department of Food and Nutrition, Meiho University, Pingtung 91202, Taiwan; 3Yu Jun Biotechnology Co., Ltd., Kaoshiung 807, Taiwan; nmm10023@yahoo.com.tw; 4Department of Pediatrics, Antai Medical Care Corporation Antai Tian-Sheng Memorial Hospital, Pingtung 92842, Taiwan; sayioung@gmail.com; 5National Museum of Marine Biology and Aquarium, Pingtung 94450, Taiwan; x2219@nmmba.gov.tw

**Keywords:** sinulariolide, soft coral, gastric cancer cells, cell migration, cell invasion

## Abstract

Cancer metastasis is the main cause of death in cancer patients; however, there is currently no effective method to predict and prevent metastasis of gastric cancer. Therefore, gaining an understanding of the molecular mechanism of tumor metastasis is important for the development of new drugs and improving the survival rate of patients who suffer from gastric cancer. Sinulariolide is an active compound isolated from the cultured soft coral *Sinularia flexibilis*. We employed sinulariolide and gastric cancer cells in experiments such as MTT, cell migration assays, cell invasion assays, and Western blotting analysis. Analysis of cell migration and invasion capabilities showed that the inhibition effects on cell metastasis and invasion increased with sinulariolide concentration in AGS and NCI-N87 cells. Immunostaining analysis showed that sinulariolide significantly reduced the protein expressions of MMP-2, MMP-9, and uPA, but the expressions of TIMP-1 and TIMP-2 were increased, while FAK, phosphorylated PI3K, phosphorylated AKT, phosphorylated mTOR, phosphorylated JNK, phosphorylated p38MAPK, and phosphorylated ERK decreased in expression with increasing sinulariolide concentration. From the results, we inferred that sinulariolide treatment in AGS and NCI-N87 cells reduced the activities of MMP-2 and MMP-9 via the FAK/PI3K/AKT/mTOR and MAPKs signaling pathways, further inhibiting the invasion and migration of these cells. Moreover, sinulariolide altered the protein expressions of E-cadherin and N-cadherin in the cytosol and Snail in the nuclei of AGS and NCI-N87 cells, which indicated that sinulariolide can avert the EMT process. These findings suggested that sinulariolide is a potential chemotherapeutic agent for development as a new drug for the treatment of gastric cancer.

## 1. Introduction

Gastric cancer is one of the most common types of cancer, and is the third leading cause of cancer death. Treatment of gastric cancer mainly involves surgery, chemotherapy, and radiation therapy. Recent improvements in diagnostic and therapeutic strategies have assisted in the early detection of cancer and reduced patient mortality [1]; however, the five-year survival rate of patients with gastric cancer is still very low. New drugs for gastric cancer treatment, such as S-1, tananes, capecitabine, oxaliplatin, and irinotecan, x * have enhanced the prognosis of patients [2,3,4,5], but the survival rate remains unsatisfactory [6,7]. At present, there is no effective method to predict and prevent gastric cancer metastasis. Therefore, gaining an understanding of the molecular mechanism of gastric cancer metastasis is important for further development of new drugs in order to improve the survival rate of patients with gastric cancer.

Marine soft corals are rich in biologically active substances that possess bioactive constituents, such as anticancer, antifungal, anti-inflammatory, antiviral, and cytotoxic materials [8]. Natural diterpenes are secondary metabolites commonly found in terrestrial and marine organisms. Furthermore, cytotoxicity is one of the main characteristics of such compounds [9,10,11]. Many researchers have reported that compounds isolated from marine soft corals, such as diterpenoids, diterpenes, and prostanoids, have an effect on apoptosis in several types of cancer cells, including melanoma, oral squamous cell carcinoma, bladder, breast, cervical, colon, and liver cancer cells [12,13,14,15,16,17,18]. Sinulariolide is an active compound extracted from cultured soft coral *Sinularia flexibilis*. Sinulariolide inhibits the proliferation and migration of the A375 melanoma cell line, the pathway of which is known to be induction of apoptosis though caspase cascade activation and mitochondrial dysfunction [9]. Sinulariolide has been shown to inhibit cell proliferation and induce apoptosis through the p38MAPK pathway in bladder cancer [10]; it has also been found to induce hepatoma cell apoptosis through a mitochondria-related pathway and an ER-stress pathway [11]. Other studies have demonstrated that sinulariolide-conjugated hyaluronan nanoparticles have anticancer effects on lung adenocarcinoma cells [12]. It has also been shown that sinulariolide reduces the release of tumor necrosis factor-α, interleukin (IL)-6, IL-12, and nitric oxide from lipopolysaccharide (LPS)-activated dendritic cells, decreasing their ability to stimulate allogeneic T cell proliferation and inhibiting the LPS-induced nuclear factor-κB pathway [13]. Recently, sinulariolide has been reported to suppress cell migration and invasion by inhibition of matrix metalloproteinase (MMP)-2/-9 through the PI3K/AKT/mTOR signaling pathway in hepatoma and bladder cancer [14,15].

However, the effects of sinulariolide on gastric cancer cells and its mechanism of action have not yet been studied. In this study, we investigated the antimigratory and anti-invasive effects of sinulariolide on human gastric cancer cell lines AGS and NCI-N87, with the aim of gaining a deeper understanding of the antimetastatic mechanisms of sinulariolide on gastric carcinoma.

## 2. Results

### 2.1. Effects of Sinulariolide Treatment on Cell Survival of Gastric Cancer Cells

The effect of sinulariolide on cell viability was assessed using an MTT assay. AGS and NCI-N87 cells were treated with sinulariolide (0–14 μM) for 24 h, and the results showed that sinulariolide had a cytotoxic effect on AGS and NCI-N87 cells at high concentrations. The cell survival rate after treatment with 10 μM sinulariolide was reduced to approximately 70%, and higher concentrations significantly decreased the cell viability to a survival rate below 60% (Figure 1). In order to avoid effects of cell migration on cell death, the highest concentration of sinulariolide used in this study was 10 μM.

### 2.2. Inhibition of Metastasis and Invasion of AGS and NCI-N87 Cells by Sinulariolide

Cell metastasis is a complex process of tumor development that includes cell migration and invasion. We used a Boyden chamber to assess the effects of sinulariolide on cell migration and invasion in AGS and NCI-N87 gastric cancer cells. Sinulariolide at concentrations of 4, 8, or 10 μM were added to cultures, and it was observed that cell migration and metastasis were reduced significantly with increased concentrations of sinulariolide (Figure 2).

### 2.3. Effects of Sinulariolide on Expressions of MMP-2, MMP-9, uPA, TIMP-1, and TIMP-2

MMP-2 and MMP-9, which are proteolytic proteins of the outer membrane, are thought to be involved in cell metastasis and cell invasion [16,17], in addition to angiogenesis. Western blotting analysis and gelatin zymography assays demonstrated that the protein expression levels and enzyme activities of MMP-2/-9 were associated with the cell migration and invasion abilities of AGS and NCI-N87 cells. The results indicated that sinulariolide inhibited the protein expressions of MMP-2, MMP-9, and uPA, and upregulated the protein expressions of TIMP-1 and TIMP-2; in addition, the enzyme activities of MMP-2 and MMP-9 were decreased (Figure 3).

### 2.4. Effects of Sinulariolide Treatment on the Intracellular FAK/PI3K/AKT/mTOR Signaling Pathway

In order to understand whether sinulariolide affects cell metastasis via the FAK/PI3K/AKT/mTOR signaling pathway, we used Western blotting to examine changes in the relevant proteins in the pathway. The results showed that sinulariolide downregulated the expressions of FAK, p-PI3K, p-AKT, p-mTOR, and RhoA in AGS and NCI-N87 cells (Figure 4).

### 2.5. Sinulariolide Inhibits MAPKs Signaling Pathways, Affecting Cell Metastasis and Invasion

Mitogen-activated protein kinases (MAPKs) are a group of serine/threonine protein kinases that have important roles in cell growth, cell differentiation, and apoptosis [18,19]. Additionally, MAPKs have been shown to be involved in metastasis [20]. Continuous activation of the JNK and p38 MAPK message pathways causes neuronal apoptosis, while ERK message pathways cause tumor neoplasia, including cancer cell proliferation, invasion, and movement [21].

In this study, Western blot analysis of sinulariolide-treated AGS and NCI-N87 cells showed that the levels of p-JNK, p-Jun, p-p38, and p-ERK were decreased with increasing sinulariolide concentration; the expression levels of metastasis-related proteins Son of sevenless homolog 1 (SOS-1), growth factor receptor-bound protein 2 (GRB2), and Ras were also reduced (Figure 5). Our results indicated that inhibition of MAPKs signaling pathways reduced cell metastasis and invasion. It was speculated that sinulariolide inhibits MAPKs signaling pathways and metastasis-related protein expressions, reducing gastric cancer cell metastasis.

### 2.6. Sinulariolide Suppressed Epithelial–Mesenchymal Transition

Epithelial cells may develop invasive mesenchymal stem cell-like properties through epithelial–mesenchymal transition (EMT). In order to establish whether sinulariolide treatment inhibits EMT in gastric cells, we employed Western blotting to determine the protein expressions of key proteins in the EMT process, including E-cadherin and N-cadherin in the cytosol, and Snail in the nucleus. The results showed that the E-cadherin protein expression was increased, while the N-cadherin expression was decreased in both AGS and NCI-N87 cells after sinulariolide treatment. Moreover, both AGS and NCI-N87 cells treated with sinulariolide exhibited reduced levels of Snail protein in the nucleus (Figure 6).

## 3. Discussion

Cancer metastasis is a major cause of cancer mortality. Researchers have found that cancer cell migration and invasion, which lead to metastasis, involve many signaling pathways [22,23]. Alteration of the integrity between cancer cells and the extracellular matrix (ECM) is an important factor leading to cancer metastasis. The ECM plays important roles in the cellular properties of cancer cells, including cell adhesion, and tumor development [24]. Cancer cells often release serine proteinase and MMPs, which can degrade ECM proteins. MMP-2 and MMP-9 are highly-expressed in malignant tumors, and have been shown to be involved in degradation of the ECM, a crucial component of the basal membrane, which consequently leads to cancer metastasis [25,26].

MMP-2 and MMP-9 can be activated through the urokinase plasminogen activator (uPA)/plasmin cascade, and both serve as important molecules in terms of facilitating invasion and metastasis of cancer cells [27]. Activation of plasmin is involved in the proteolytic cleavage of the Arg560Val562 peptide bond of plasminogen by uPA. This process triggers a series of reactions, including activation of MMPs, causing degradation of several types of collagen, leading to tumor metastasis. Additionally, uPA is known to be associated with cancer cell proliferation, cell invasion, and angiogenesis [28]. MMP-2, MMP-9, and uPA are thought to play important roles in the degradation of ECM proteins, which is a key factor in tumor metastasis and invasion [29], and inhibition of their protein expressions and proteolytic activities can be an important strategy to prevent tumor metastasis and invasion. Altered expressions of protease inhibitors are known to be involved in many pathological processes, including tumorigenesis, inflammation, and angiogenesis. Tissue inhibitors of metalloproteinases (TIMPs) are general endopeptidase inhibitors that inhibit the activities of MMPs [30]. Maintaining a balance between TIMPs and MMPs is important, as destruction of this homeostatic balance may cause ECM degradation and accelerate metastasis of cancer [16,17]. As shown by the results of this study, sinulariolide suppressed the protein expressions of uPA, MMP-2, and MMP-9, and increased the protein expression levels of TIMP-1/-2. These findings were in agreement with another report indicating that sinulariolide suppressed cell migration and invasion in cancer cells [14,15].

FAK is associated with the proliferation and metastasis of cancer cells [31]. The FAK signaling pathway can regulate the expression of MMP-2, which inhibits the invasion of head and neck squamous cell carcinoma [32]. The PI3K/AKT/mTOR, Ras/Raf/MAPKs, and NF-κB signaling pathways have been shown to drive oncogenesis in cancer. Studies have demonstrated that suppression of the PI3k/AKT/mTOR pathway by an mTOR inhibitor may inhibit cancer cell invasion and migration and promote apoptosis in tumors, while PI3K signaling is known to contribute to the advancement of cancer [33,34,35,36]. Overexpression of KiSS-1 reduces the invasiveness of colorectal cancer cells by blocking the PI3K/AKT/NF-κB pathway and inhibiting the expression of MMP-9 protein [37]. Studies have shown that RhoA signaling and the PI3K/AKT pathway mediate cancer cell invasion [38,39], and the FAK–RhoA pathway may influence cancer cell motility [40,41]. According to our results, sinulariolide inhibited the FAK/PI3K/AKT/mTOR pathway and reduced the MMP-2/-9 and uPA protein expressions, preventing invasion and migration of AGS and NCI-N87 cells.

SOS-1 is an important epidermal growth factor receptor (EGFR) [42], and growth factor receptor-bound protein 2 (GRB2) is an adaptor protein involved in signal transduction pathways [43]. SOS-1 can be recruited through GRB2 and activates receptors or scaffold proteins on the membrane, subsequently activating Ras [42]; Ras then affects the phosphorylation of p38MAPK, JNK, and ERK [44]. The results of Western blot analysis also showed that the levels of p-JNK, p-p38MAPK, and p-ERK were decreased in AGS and NCI-N87 cells with increasing sinulariolide concentration, and the expression levels of metastasis-related proteins SOS-1 and Ras were also reduced.

The MAPKs signaling pathway is important for anticancer therapy [18,45]. Involvement of intracellular signaling pathways has been shown to be required in many studies of cell metastasis and invasion in liver cancer. Currently, the ERK, p38MAPK, and JNK signaling pathways are considered to be involved [46,47,48,49]. In this study, it was found that sinulariolide prevented the invasion and migration of AGS and NCI-N87 cells by inhibiting the JNK/p38MAPK/ERK pathway and reducing the expressions of MMP-2/-9 and uPA.

In order to identify the mechanisms involved in sinulariolide-mediated inhibition of cell migration and invasion in AGS and NCI-N87 cells, we analyzed proteins associated with EMT and metastasis. E-cadherin loses its function when cancer cells metastasize, causing the cells to transfer to other organs via blood circulation [50]. Downregulated E-cadherin and upregulated N-cadherin expressions are crucial in the EMT process, inducing cell adhesion to stroma and augmenting the invasiveness of tumor cells during cancer metastasis [51,52]. Snail is one of the most important molecules in the EMT process; it binds to the E-box region of the E-cadherin promoter, directly inhibiting E-cadherin DNA transcription, reducing its protein expression, and promoting EMT [53]. Snail also affects the ability of cancer cells to invade the surrounding tissue [54]. In the EMT process, loss of cell adhesion ability is accompanied by changes in molecular type, including decreased expressions of structural proteins in epithelial cells and increased expressions of structural proteins in mesenchymal cells. The results of this study showed that in sinulariolide-treated cells, the E-cadherin level was increased, while the Snail and N-cadherin expressions were decreased, indicating that sinulariolide can prevent the EMT process.

## 4. Materials and Methods

### 4.1. Cell Culture, Drug Treatment, and Cell Viability Assay

Human gastric cancer AGS (human gastric cancer epithelial cell line) and NCI-N87 (a gastric cancer cell line which metastatic from liver site) cell lines were obtained from the Taiwan Food Industry Research and Development Institute (Hsinchu, Taiwan). The cell lines were cultured in Dulbecco’s modified Eagle’s medium (DMEM) supplemented with 10% fetal bovine serum (FBS), 100 µg/mL streptomycin and 100 ug/mL penicillin in a humidified 5% CO_2_ incubator at 37 °C. The cell viability assay was performed based on the method described in a previous study [55]. AGS and NCI-N87 cells (1 x 10^5^/well) were seeded in 96-well plates and treated with sinulariolide at final concentrations of 2–14 µM. After 24 h of incubation, MTT solution was added to each well, and the absorbance was measured at 595 nm using a microplate ELISA reader (Bio-Rad, Hercules, CA, USA). DMSO treatment was used as the control. All experiments were performed in triplicate to determine their reproducibility.

### 4.2. Cell Migration and Invasion Analysis by Transwell Assay

The Transwell assay was performed as described in a previous study [56]. AGS and NCI-N87 cells in serum-free media were placed in an uncoated Boyden chamber (Neuro Probe, Cabin John, MD, USA) at 5 × 10^4^ cells/well, and cultured with or without sinulariolide treatment in a 37 °C CO_2_ incubator for 24 h to allow cells to migrate through the membrane. Transwell inserts with a collagen coating (8 m pore size; BD Biosciences, Franklin Lakes, NJ, USA) were used, and AGS and NCI-N87 cells were imported onto the coated membrane in the upper chamber, while cell culture medium containing 10% FBS was placed in the lower chamber. After incubation, cells invaded through the Matrigel-coated membrane to the lower chamber. These cells were then fixed with ice-cold methanol, followed by staining with Giemsa solution (concentration = 5%; Merck, Darmstadt, Germany), and counted under a light microscope.

### 4.3. Gelatin Zymography Assay

A gelatin zymography assay was employed to determine the proteolytic activities of MMP-2/-9. The experimental protocol was according to Yang’s study [56]. AGS and NCI-N87 cells were treated with sinulariolide at concentrations of 2, 4, 8, and 10 M for 24 h. Samples of conditional media were collected and concentrated using a vacuum concentrator. The samples were then separated on SDS-PAGE (10%) containing 0.2% gelatin under non-reducing conditions and washed three times in 100 mM NaCl/50 mM Tris-HCl (pH = 7.5) buffer containing 2.5% Triton-X-100. The enzymes in the gels were incubated at 37 °C for 24 h in reaction buffer (200 mM NaCl, 0.02% NaN_3_, 1 M ZnCl_2_, 1 mM CaCl_2_, and 2% Triton-X 100 in 50 mM Tris-HCl buffer, pH 7.5). The gels were then stained with Coomassie blue R-250 dye, and thereafter de-stained.

### 4.4. Protein Concentration Analysis and Western Blot Assay

AGS and NCI-N87 cells (3 × 10^5^ cells) were seeded into 10 cm diameter dishes and incubated in FBS-DMEM media with various concentrations of sinulariolide (0, 2, 4, 8, and 10 M) for 24 h. After incubation, the cells were lysed with a cell extraction buffer (BioSource International, Camarillo, CA, USA) and the protein concentrations were determined using a Bradford protein assay (Bio-Rad). From the samples treated with sinulariolide or DMSO (as the control), total proteins (25 µg) were separated by SDS-PAGE (12.5%), then transferred to a PVDF membrane. The membrane containing transferred proteins was then blocked in phosphate-buffered saline (PBS) containing 5% low-fat milk powder in order to eliminate nonspecific binding. The membrane was then blotted with primary antibodies at 4 °C overnight, and subsequently incubated with secondary antibodies (dilution = 1/5000 in blocking solution) at 4 °C for 2 h. The immunoreactive protein concentrations were measured using the chemiluminescence substrate solution (Pierce Biotechnology, Rockford, IL, USA). Rabbit anti-human PI3K, p-PI3K, uPA, Lamin A2, E-cadherin, N-cadherin, and Snail antibodies were obtained from ProteinTech Group (Chicago, IL, USA); rabbit anti-human SOS-1, FAK, TIMP-1, TIMP-2, MMP-9, p38, p-p38, mTOR, p-mTOR, GRB2, AKT, p-AKT, and-actin antibodies were obtained from Cell Signaling Technology (Danvers, MA, USA); and rabbit anti-human Jun, p-Jun, ERK, p-ERK, JNK, p-JNK, and Ras antibodies were obtained from Epitomics (Burlingame, CA, USA). The Western blot data were quantified with the ImageJ 1.47 software (National Institutes of Health, Bethesda, MD, USA).

### 4.5. Statistical Analysis

MTT, cell migration, and invasion assay data were collected from three independent experiments and analyzed using Student’s t-test (Sigma-Stat2.0, San Rafael, CA, USA). Results with *p* < 0.05 were considered statistically significant.

## 5. Conclusions

In the present study, we proved that sinulariolide has a substantial effect in terms of reduction of gastric cancer cell migration and invasion, and also halts the EMT process, in AGS and NCI-N87 cells. Our results also showed that signaling pathways associated with MMP-2, -9, and uPA regulation are vital in gastric cancer cells, and sinulariolide efficiently blocks gastric cancer metastasis via targeting the signaling pathways driving the processes of gastric cancer advancement (Figure 7). Our findings indicated that sinulariolide is a potential agent for development as a new drug for the treatment of gastric cancer.

## Figures and Tables

**Figure 1 marinedrugs-17-00668-f001:**
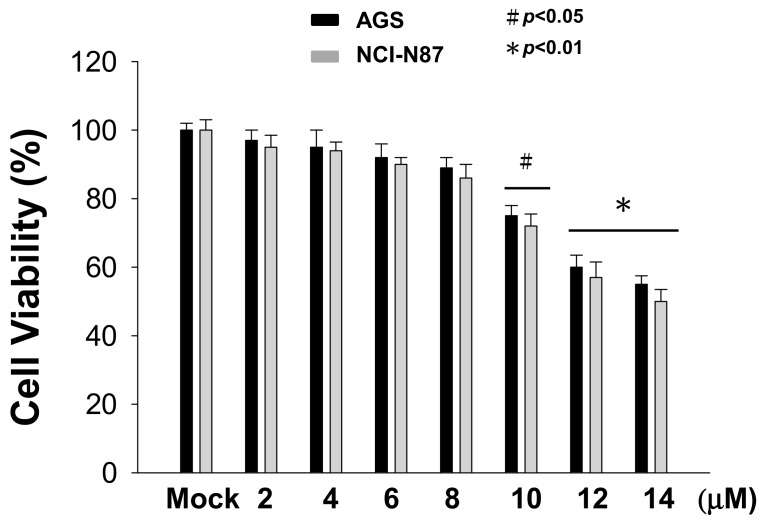
High concentrations of sinulariolide decreased the cell viabilities of AGS and NCI-N87 cells. AGS and NCI-N87 cells were treated with different concentrations of sinulariolide for 24 h. MTT assays showed cytotoxic effects of sinulariolide treatment on AGS and NCI-N87 cells in a concentration-dependent manner (# *p* < 0.05; * *p* < 0.01 compared with the controls). The results were obtained from three individual experiments.

**Figure 2 marinedrugs-17-00668-f002:**
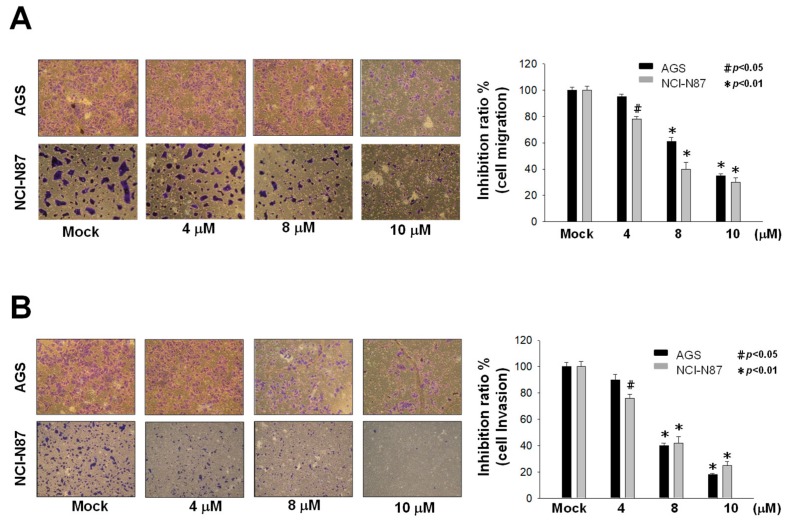
Sinulariolide suppressed cell migration and invasion of AGS and NCI-N87 cells. After 24 h of treatment with sinulariolide, the percentages of AGS and NCI-N87 cells that had migrated were significantly lower compared with the controls (cultures treated with DMSO vehicle control (Mock)). The results were quantitated in three independent experiments. Dose-dependent inhibition effects of sinulariolide on (**A**) migration and (**B**) invasion of AGS and NCI-N87 cells were observed (# *p* < 0.05, * *p* < 0.01 compared with the controls). The results were obtained from three individual experiments.

**Figure 3 marinedrugs-17-00668-f003:**
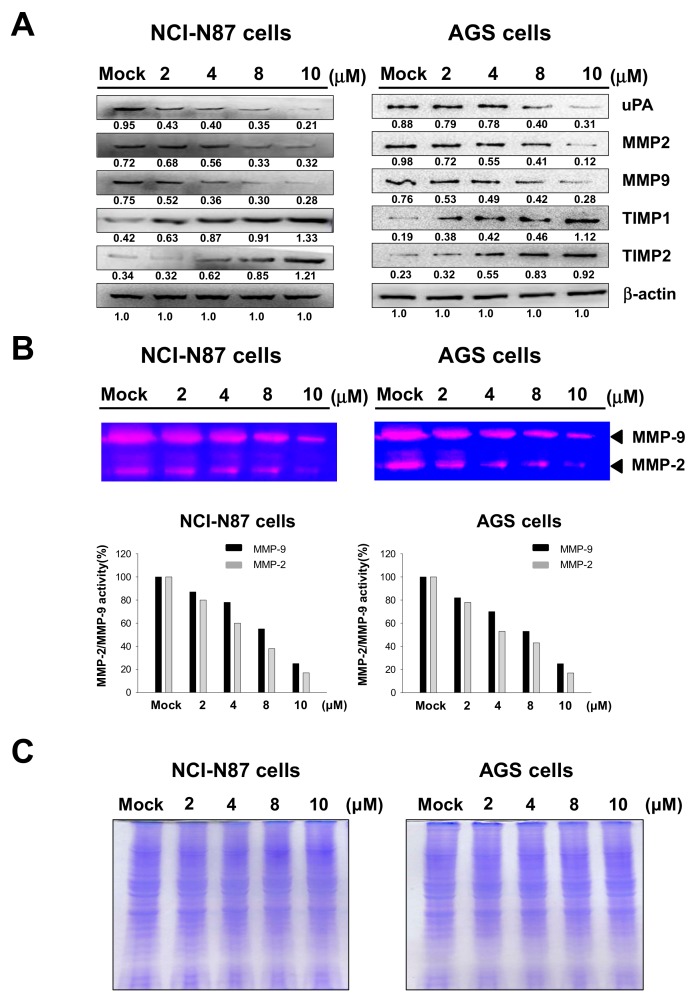
Sinulariolide regulated the expressions and proteolytic activities of matrix metalloproteinases (MMP)-2 and MMP-9 and their inhibitors in AGS and NCI-N87 cells. (**a**) Western blot analysis of the protein expression levels of urokinase plasminogen activator (uPA), MMP-2, MMP-9, tissue inhibitors of metalloproteinases (TIMP)-1, and TIMP-2 in cells treated with different concentrations of sinulariolide. β-actin was used as the loading control. (**b**) Gelatin zymography showed that sinulariolide inhibited the proteolytic activities of MMP-2 and MMP-9 in AGS and NCI-N87 cells. Mock: Cells treated with vehicle control (DMSO). Quantification of MMP-2 and MMP-9 using ImageJ 1.47 software (National Institutes of Health, Bethesda, MD, USA). (**c**) SDS-PAGE gel stained with Coomassie stain was used as the zymography gels proteins loading control.

**Figure 4 marinedrugs-17-00668-f004:**
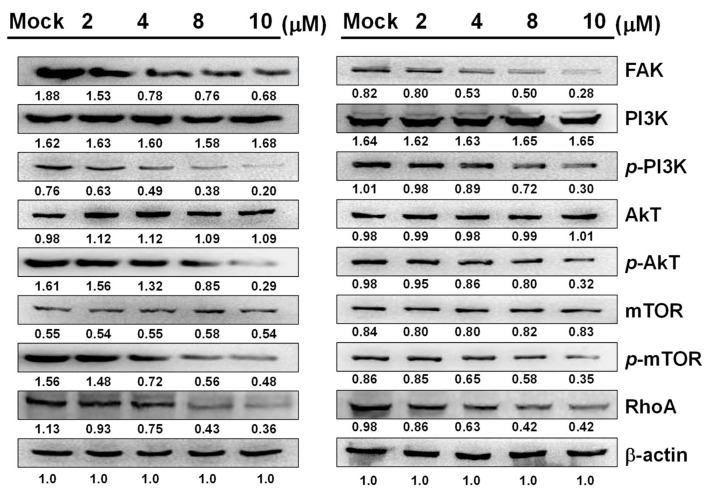
Sinulariolide regulated the FAK/PI3K/AKT/mTOR signaling pathway in AGS and NCI-N87 cells, inhibiting the protein expressions of FAK, p-PI3K, p-AKT, p-mTOR, and RhoA. β-actin was used as the loading control.

**Figure 5 marinedrugs-17-00668-f005:**
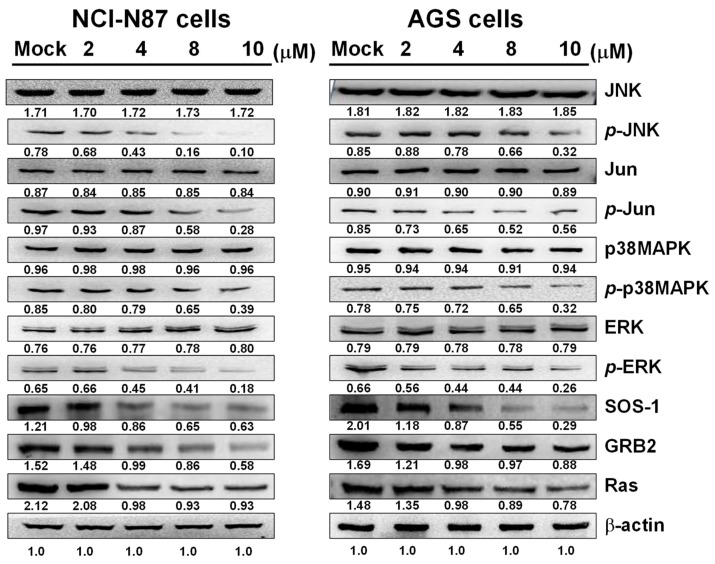
Sinulariolide regulated mitogen-activated protein kinases (MAPKs) signaling pathways and metastasis-related proteins in AGS and NCI-N87 cells, reducing the protein expressions of p-JNK, p-Jun, p-p38, p-ERK, Son of sevenless homolog 1 (SOS-1), growth factor receptor-bound protein 2 (GRB2), and Ras. Mock: Cells treated with vehicle control (DMSO). β-actin was used as the loading control.

**Figure 6 marinedrugs-17-00668-f006:**
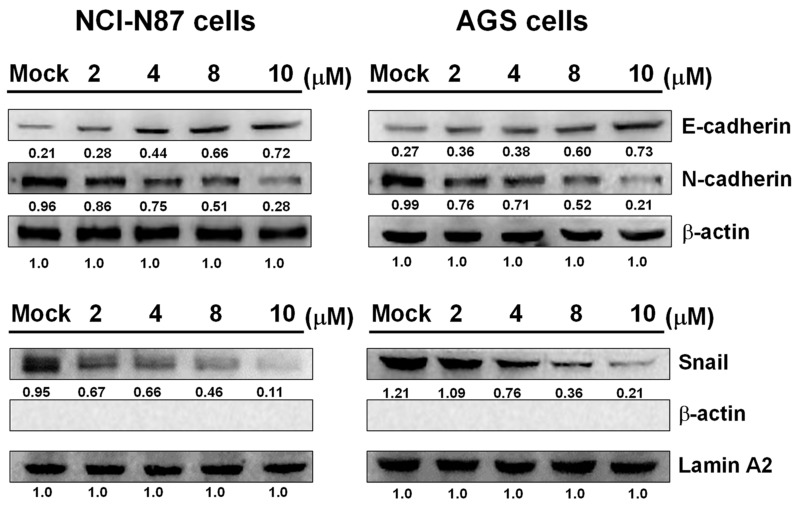
Sinulariolide inhibited the epithelial to mesenchymal transition (EMT) process in AGS and NCI-N87 cells. Cells were treated with various concentrations of sinulariolide (2, 4, 8, and 10 µM), and the expressions of EMT-associated proteins E-cadherin, N-cadherin, and Snail were quantitated by Western blotting. Sinulariolide treatment downregulated Snail and N-cadherin expressions, while increasing the E-cadherin expression, in AGS and NCI-N87 cells. β-actin and Lamin A2 were used as the internal controls for cytosol and nucleus proteins, respectively. Mock: Cells treated with vehicle control (DMSO).

**Figure 7 marinedrugs-17-00668-f007:**
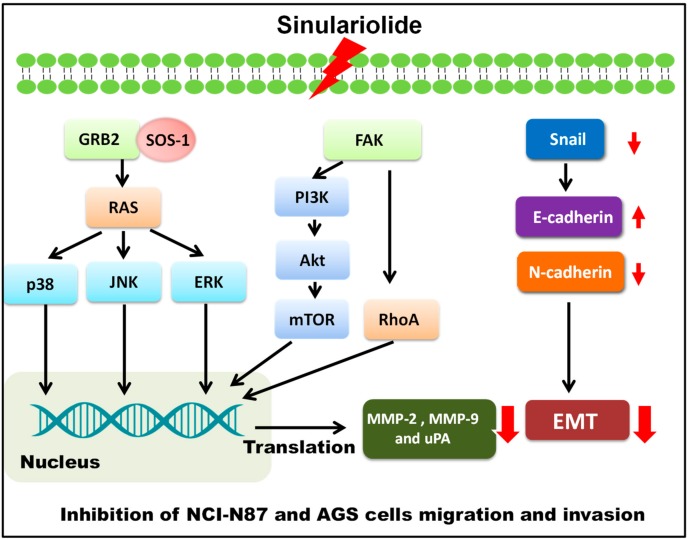
Proposed signaling pathways for sinulariolide-mediated inhibition of gastric cancer cell migration and invasion.

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
