# Peer review of "Sinulariolide Inhibits Gastric Cancer Cell Migration and Invasion through Downregulation of the EMT Process and Suppression of FAK/PI3K/AKT/mTOR and MAPKs Signaling Pathways"

_marinedrugs, 2019, doi:10.3390/md17120668_

Round 1

Reviewer 1 Report

Summary

            In the article titled “Sinulariolide inhibits gastric cancer cell migration and invasion through downregulation of the EMT process…” the authors show that cell migration and EMT is reduced in two gastric cancer cell lines upon treatment with sinulariolide, a compound extracted from a soft marine coral. The authors extensively test the expression of several components of various signaling pathways, including FAK, PI3K, AKT and mTOR in addition to examining the expression of cancer associated matrix metalloproteases. While their experimental methods are sound, some additional data analysis should be performed in order to strengthen their conclusions, including quantifying the western blot data that is critical to their stated conclusions. Furthermore, some additional explanation of some of their methods is needed. The following major and minor issues should be addressed before this manuscript should be accepted for publication.

Major Issues

The authors use two cell lines throughout their experiments, but do not provide any justification for the use of these cell lines, nor do they appear to test the effects of this compound on a normal, non-cancerous cell line. What is different between the AGS and NCI-N87 cell lines? Is one more aggressive than the other? It would be helpful to test a normal cell line (for baseline comparison), a mildly aggressive and a highly aggressive metastatic cell line to see if there is a graduated effect of this compound on metastatic readout (migration, invasion, EMT, etc.). On a similar line of thought with the previous issue, in Figure 2, there appears to be a switch between the effect on the two cell lines when treated with the drug in terms of migration vs invasion. For example, the NCI-N87 cell line responds more greatly in the migration assay (Figure 2A) but the other cell line (AGS) responds more so in the invasion assay (Figure 2B). The authors should speculate on this slight, yet interesting swap in phenotype. It may lend to differences between the two cell lines, which should be addressed as raised in the above point. For all of the Western blot data, there is no quantification provided. While I realize this isn’t needed for all of the data, as some only contribute minor detail to the overall story, but some of the changes in expression is critical to the story, such as the MMP data and the FAK levels (especially since the two cell lines appear to have varying background levels of FAK expression before drug treatment). The authors should provide quantification of expression levels for critical parts of their conclusions. It would help to make their arguments more convincing. The authors should test for the level of active RhoA kinase, not only total RhoA (as they have already done). This can be done by a pull down of active RhoA by Rhotekin 7-89 bound beads. For the zymography gels performed in Figure 3, there needs to be some way to know that these gels were equally loaded. The gels should be run in parallel with another SDS-PAGE gel that can be stained with Coomassie stain so that the reader can see that the lanes are equally loaded.

Minor Issues

A careful edit for English grammar would be helpful as there are many errors in the manuscript’s current form that detract from the overall impact of the conclusions. In the section regarding EMT transition, do the cells exhibit a change in general morphology? It would be helpful to have some microscopy images if there is a change in morphology. For all experiments, there appears to be no mention of the number of repeats/biological replicates that were performed.

Author Response

Reviewer 1

 Summary

            In the article titled “Sinulariolide inhibits gastric cancer cell migration and invasion through downregulation of the EMT process…” the authors show that cell migration and EMT is reduced in two gastric cancer cell lines upon treatment with sinulariolide, a compound extracted from a soft marine coral. The authors extensively test the expression of several components of various signaling pathways, including FAK, PI3K, AKT and mTOR in addition to examining the expression of cancer associated matrix metalloproteases. While their experimental methods are sound, some additional data analysis should be performed in order to strengthen their conclusions, including quantifying the western blot data that is critical to their stated conclusions. Furthermore, some additional explanation of some of their methods is needed. The following major and minor issues should be addressed before this manuscript should be accepted for publication.

Major Issues

1.The authors use two cell lines throughout their experiments, but do not provide any justification for the use of these cell lines, nor do they appear to test the effects of this compound on a normal, non-cancerous cell line. What is different between the AGS and NCI-N87 cell lines? Is one more aggressive than the other? It would be helpful to test a normal cell line (for baseline comparison), a mildly aggressive and a highly aggressive metastatic cell line to see if there is a graduated effect of this compound on metastatic readout (migration, invasion, EMT, etc.).

Responds:

In order to observe the effect of sinulariolide on cell migration and invasion of gastric cancer, we employed human gastric cancer epithelial cell line AGS and NCI-N87 which is mastasize from liver site. We have added the information in material methods.  

The concentrations of sinulariolide in this study have been examined in normal skin cell (HaCat cells). The results (the figure shown as below) exhibited that the toxicity of sinulariolide is obviously lower in HeCat cells.

On a similar line of thought with the previous issue, in Figure 2, there appears to be a switch between the effect on the two cell lines when treated with the drug in terms of migration vs invasion. For example, the NCI-N87 cell line responds more greatly in the migration assay (Figure 2A) but the other cell line (AGS) responds more so in the invasion assay (Figure 2B). The authors should speculate on this slight, yet interesting swap in phenotype. It may lend to differences between the two cell lines, which should be addressed as raised in the above point.

Responds:

In the experiment, the two cells (AGS and NCI-N87 cells) were cultured in the same number of cells ( 5×104). It was found by migration and invasion that NCI-N87 cells were more sensitive to sinulariolide. Therefore, when NCI-N87 cell was treated by sinulariolide, the number of cells in migration and invasion was relatively less than AGS.

For all of the Western blot data, there is no quantification provided. While I realize this isn’t needed for all of the data, as some only contribute minor detail to the overall story, but some of the changes in expression is critical to the story, such as the MMP data and the FAK levels (especially since the two cell lines appear to have varying background levels of FAK expression before drug treatment). The authors should provide quantification of expression levels for critical parts of their conclusions. It would help to make their arguments more convincing.

Responds: Thank for reviewer’s suggestion. All the Western blot data were quantified with Image J 1.47 software

4.The authors should test for the level of active RhoA kinase, not only total RhoA (as they have already done). This can be done by a pull down of active RhoA by Rhotekin 7-89 bound beads.

Responds: Thank for reviewer suggestion. We have not this reagent for this experiment in laboratory. If there is any relevant experiment in the future, the reviewer’s suggestion will use the valuable opinions to conduct research.

 5.For the zymography gels performed in Figure 3, there needs to be some way to know that these gels were equally loaded. The gels should be run in parallel with another SDS-PAGE gel that can be stained with Coomassie stain so that the reader can see that the lanes are equally loaded.

Responds: 

Thank for reviewer’s suggestion. We have re-made SDS-PAGE gel in figure 3 C.

Minor Issues

A careful edit for English grammar would be helpful as there are many errors in the manuscript’s current form that detract from the overall impact of the conclusions. In the section regarding EMT transition, do the cells exhibit a change in general morphology? It would be helpful to have some microscopy images if there is a change in morphology. For all experiments, there appears to be no mention of the number of repeats/biological replicates that were performed.

Responds: Thank for reviewer’s suggestion. We have modified the English grammar. We have mention of the number of repeats/biological replicates in in figure 1-2.

The morphology of these two cell lines showed no significant changes.

Reviewer 2 Report

Dear authors,

before any comment about the scientific content, I recommend extensive editing of English language and style.

I could nor read the paper! English language is so poor that the meaning of some sentences is completely lost! 

Examples;

Abstract

Sinulariolide is an active compound that isolated from the cultured soft coral Sinularia flexibilis

Cell migration and invasion ability analysis showed that the inhibit ability that to cell metastasis and invasion increased with sinulariolide concentration.

It revealed that the effect of inhibition the migration...

Moreover, sinulariolide altering the protein expression included E-cadherin,...

Introduction

Recently improvements in diagnostic and therapeutic strategies...

Therefore, understanding the molecular mechanism of gastric cancer metastasis is important for further to develop of new drugs improve the survival rate...

Sinulariolide inhibits the proliferation 58 and migration of the A375 melanoma cell line, the pathway of which is known to be induction...

Materials and methods

The cells were treated with sinulariolide at the final concentrations were 2-14 μM respectively.

The methods were according to previously study.

AGS and NCI-N87 cells in serum free media and were plated into an uncoated...

AGS and NCI-N87 cells with or without sinulariolide treatment and cultured in 252 a 37℃ CO2 incubator for 24 h let cells to migrate through the membrane.

Transwell inserts which with collagen coating..

AGS 254 and NCI-N87 cells were import onto the coated membrane where upper chamber. In the lower 255 chamber, cell culture medium containing 10% FBS was placed.

Gelatin zymography assays were employed to determine MMP-2/-9 and all The experiment protocol was according to Yang’s study [56].

I stopped to read here. I can not understand what have you been done... I can only guess.. 

Please, review whole paper for accuracy of scientific style, writing and grammar.

Author Response

Reviewer 2:

Reviewer’s Comments

Authors’ Responses

Dear authors,

before any comment about the scientific content, I recommend extensive editing of English language and style.

I could nor read the paper! English language is so poor that the meaning of some sentences is completely lost!

We are sorry for the poor English writing. Our article have be English edited by professional English editor.

The sentence has been revised as follows:

Abstract section

Sinulariolide is an active compound that isolated from the cultured soft coral Sinularia flexibilis

Sinulariolide is an active compound isolated from the cultured soft coral Sinularia flexibilis.

Cell migration and invasion ability analysis showed that the inhibit ability that to cell metastasis and invasion increased with sinulariolide concentration.

Analysis of cell migration and invasion capabilities showed that the inhibition effects on cell metastasis and invasion increased with sinulariolide concentration in AGS and NCI-N87 cells.

It revealed that the effect of inhibition the migration...

Moreover, sinulariolide altering the protein expression included E-cadherin,..

Moreover, sinulariolide altered the protein expressions of E-cadherin and N-cadherin in the cytosol and Snail in the nuclei of AGS and NCI-N87 cells,

Introduction section

Recently improvements in diagnostic and therapeutic strategies...

Recent improvements in diagnostic and therapeutic strategies have assisted in the early detection of cancer and reduced patient mortality [1]

Therefore, understanding the molecular mechanism of gastric cancer metastasis is important for further to develop of new drugs improve the survival rate...

Therefore, gaining an understanding of the molecular mechanism of gastric cancer metastasis is important for further to development of new drugs in order to improve the survival rate of patients which suffer from with gastric cancer

Sinulariolide inhibits the proliferation 58 and migration of the A375 melanoma cell line, the pathway of which is known to be induction...

Sinulariolide inhibits the proliferation and migration of the A375 melanoma cell line, the pathway of which is known to be induction of apoptosis though caspase cascade activation and mitochondrial dysfunction [9].

Materials and methods section

The cells were treated with sinulariolide at the final concentrations were 2-14 μM respectively.

AGS and NCI-N87 cells (1x105/well) were seeded in 96-well plates and treated with sinulariolide at final concentrations of 2–14 µM.

The methods were according to previously study.

The Transwell assay was performed as described in a previous study

AGS and NCI-N87 cells in serum free media and were plated into an uncoated...

AGS and NCI-N87 cells with or without sinulariolide treatment and cultured in 252 a 37℃ CO2 incubator for 24 h let cells to migrate through the membrane.

AGS and NCI-N87 cells in serum-free media were placed in an uncoated Boyden chamber (Neuro Probe, Cabin John, MD, USA) at 5×104 cells per well, and cultured with or without sinulariolide treatment in a 37°C CO2 incubator for 24 h to allow cells to migrate through the membrane.

Transwell inserts which with collagen coating..

AGS 254 and NCI-N87 cells were import onto the coated membrane where upper chamber. In the lower 255 chamber, cell culture medium containing 10% FBS was placed.

Transwell inserts with a collagen coating (8 mm pore size, BD Biosciences, Franklin Lakes, NJ, USA) were used, and AGS and NCI-N87 cells were imported onto the coated membrane in the upper chamber, while cell culture medium containing 10% FBS was placed in the lower chamber.

Gelatin zymography assays were employed to determine MMP-2/-9 and all The experiment protocol was according to Yang’s study [56].

A gelatin zymography assay was employed to determine the proteolytic activities of MMP-2/-9. The experimental protocol was according to Yang’s study [56].

I stopped to read here. I can not understand what have you been done... I can only guess.. 

Please, review whole paper for accuracy of scientific style, writing and grammar.

Round 2

Reviewer 1 Report

The authors have satisfied my initial concerns raised in the first round of review. They have also substantially improved the English grammar of the paper. I recommend that this paper be accepted for publication.